# Peptide Dendrimers with Non-Symmetric Bola Structure Exert Long Term Effect on Glioblastoma and Neuroblastoma Cell Lines

**DOI:** 10.3390/biom11030435

**Published:** 2021-03-15

**Authors:** Marta Sowińska, Monika Szeliga, Maja Morawiak, Elżbieta Ziemińska, Barbara Zabłocka, Zofia Urbańczyk-Lipkowska

**Affiliations:** 1Institute of Organic Chemistry PAS, 01-224 Warsaw, Poland; marta.sowinska@ryvu.com (M.S.); maja.morawiak@icho.edu.pl (M.M.); 2Mossakowski Medical Research Institute PAS, 02-106 Warsaw, Poland; elziem@imdik.pan.pl (E.Z.); bzablocka@imdik.pan.pl (B.Z.)

**Keywords:** dendrimers, bola structure, glioblastoma, neuroblastoma, proliferation, colony formation assay, reactive oxygen species

## Abstract

Background: Glioblastoma (GBM) is the most common malignant tumor of the central nervous system (CNS). Neuroblastoma (NB) is one of the most common cancers of childhood derived from the neural crest cells. The survival rate for patients with GBM and high-risk NB is poor; therefore, novel therapeutic approaches are needed. Increasing evidence suggests a dual role of redox-active compounds in both tumorigenesis and cancer treatment. Therefore, in this study, polyfunctional peptide-based dendrimeric molecules of the bola structure carrying residues with antiproliferative potential on one side and the antioxidant residues on the other side were designed. Methods: We synthesized non-symmetric bola dendrimers and assessed their radical scavenging potency as well as redox capability. The influence of dendrimers on viability of rat primary cerebellar neurons (CGC) and normal human astrocytes (NHA) was determined by propidium iodide staining and cell counting. Cytotoxicity against human GBM cell lines, T98G and LN229, and NB cell line SH-SY5Y was assessed by cell counting and colony forming assay. Results: Testing of CGC and NHA viability allowed to establish a range of optimal dendrimers structure and concentration for further evaluation of their impact on two human GBM and one human NB cell lines. According to ABTS, DPPH, FRAP, and CUPRAC antioxidant tests, the most toxic for normal cells were dendrimers with high charge and an excess of antioxidant residues (Trp and PABA) on both sides of the bola structure. At 5 μM concentration, most of the tested dendrimers neither reduced rat CGC viability below 50–40%, nor harmed human neurons (NHA). The same dose of compounds **16** or **22**, after 30 min treatment decreased the number of SH-SY5Y and LN229 cells, but did not affect the number of T98G cells 48 h post treatment. However, either compound significantly reduced the number of colonies formed by SH-SY5Y, LN229, and T98G cells measured 14 days after treatment. Conclusions: Peptide dendrimers with non-symmetric bola structure are excellent scaffolds for design of molecules with pro/antioxidant functionality. Design of molecules with an excess of positive charges and antioxidant residues rendered molecules with high neurotoxicity. Single, 30 min exposition of the GBM and NB cell lines to the selected bola dendrimers significantly suppressed their clonogenic potential

## 1. Introduction

Despite significant progress that has been made recently in development of therapeutic strategies in cancer therapy, glioblastoma (GBM) remains the most common malignant tumor of the central nervous system (CNS) with five-year relative survival of 6.8% [1]. Neuroblastoma (NB) is one of the most common extracranial solid tumors in children derived from precursor cells of the peripheral nervous system (PNS). The five-year survival rate of patients with high-risk NB is below 50% [2]. Therefore, novel therapeutic approaches are needed, and, in this context, a search for selective lead compounds for either GBM or NB represents a promising approach.

Since introduction of a new lead compounds is rather seldom worldwide, the rapidly growing field of nanomedicine offers new solutions to improve the pharmacodynamics and pharmacokinetics of the existing drugs and, at medical level, the effectiveness of chemotherapy [3]. This involves development of various nanocarriers [4,5,6,7] or combining both therapeutic and diagnostic capability in the theranostic agents [8,9]. Another recent trend is the design of nanomolecules with intrinsic anticancer activity, often associated with the presence of chelated metals cations [10,11,12].

We have previously demonstrated that the second generation poly-lysine dendrimers terminated with 2-chlorobenzyloxycarbonyl (2-Cl-Z) residue inhibit the proliferation of GBM cell lines without expressing significant toxicity against non-tumoural CNS cells (neurons and glia) [13]. Cytotoxic effect was potentiated when dendrimers contained indole residue at C-terminal position. On the other hand, peptide dendrimers functionalized with *p*-aminobenzoic acid (PABA) expressed activity against human melanoma cell line, playing also protective role in neurons [14]. Therefore, we hypothesized that lower neurotoxicity of the latter was associated with antioxidant and reactive oxygen species (ROS) scavenging properties that are known attributes of PABA and indole moieties [15,16]. Possibly, that antiproliferative activity of both groups was related to the improved intracellular penetration afforded by net positive charge and amphipathic structure of dendrimers.

Accumulating evidence suggests that, in addition to the positive effect on cell wellbeing, the redox-active compounds may either promote or suppress tumorigenesis. Therefore, the aim of the present study was to design polyfunctional dendrimeric molecules with chemical structure that enables introduction of residues with previously documented antimicrobial [17], antifungal [18], or antiproliferative [13,14] activity and functions yielding antioxidant properties [14]. Chemical design was focused on structure of dendrimers with so called bola-architecture. The synthesis involved independent synthesis of two dendrons containing either cytotoxic or antioxidant moieties and coupling them together with linkers of various length and structure.

The obtained dendrimers were pre-screened for antioxidant properties and in vitro cell toxicity. In general, antioxidant activity is expressed by molecules with chemical structure that allows to accept electron(s) and transform themselves into more stable and less reactive radicals. Therefore, these compounds were primarily tested for molecular non-enzymatic antioxidant capacity by applying FRAP, DPPH, ABTS, and CUPRAC laboratory tests. Since reactive oxygen species (ROS) may work as a tumor-promoting or a tumor-suppressing factor, production of ROS in rat primary cerebellar granule cells (CGC) upon exposure to the tested compounds was measured. The structurally relevant group of dendrimers was selected for further testing for their cytotoxic activity against common human NB SH-SY5Y and GBM cell lines. Since inter- and intra-tumor heterogeneity is one of the most important hallmarks of GBM [19], two GBM cell lines, T98G and LN229, varying with respect to tumorigenic potential and molecular alterations often found in GBM [20] were used. The key premise is that potential anticancer drug should display cytotoxicity in tumor cells without affecting normal tissues; therefore, we also examined the influence of dendrimers on normal human primary astrocytes (NHA) and rat primary cerebellar neurons (CGC).

## 2. Materials and Methods

### 2.1. General Procedures

All solvents and reagents were of analytical grade and were used without further purification. All solvents were obtained from Sigma-Aldrich (Steinheim, Germany). Mass spectra were recorded with a Mariner ESI time-of-flight mass spectrometer (PerSeptive Biosystems, Foster City, CA, USA) for the samples prepared in MeOH. The ^1^H-NMR and ^13^C-NMR spectra were recorded using a Bruker Avance spectrometer (Karlsruhe, Germany) at 500/125 or 400/100 MHz, respectively, using deuterated solvents and TMS as an internal standard. Chemical shifts are reported as δ values in parts per million, and coupling constants are given in hertz. The optical rotations (**[α]_D_^25^**) were measured with JASCO J-1020 digital polarimeter (Ishikawa-machi, Hachioji, Tokyo, Japan). Melting points were recorded on a Köfler hot-stage apparatus (Wagner and Munz, München, Germany) and are uncorrected. Thin layer chromatography (TLC) was performed on aluminum sheets with silica gel 60 F254 from Merck (Darmstadt, Germany). Column chromatography (CC) was carried out using silica gel (230–400 mesh) from Merck or Sephadex LH20 (Biosciences, Upsala, Sweden). The TLC spots were visualized by treatment with 1% alcoholic solution of ninhydrin and heating.

### 2.2. Synthesis and Characterization of Bola Dendrimers

The general strategy for the design of the studied series of the unsymmetrical bola-type dendrimers involved independent synthesis of the larger left side peptide dendrons functionalized at C-terminus with the appropriate linker, followed by coupling with smaller organic right side dendrons. The synthesis of Boc-protected right side organic dendrons functionalized with two p-aminobenzoic (PABA, **7**) or p-amino-benzenosulfonic acid (PAS, **8**) moieties is shown in Scheme 1. Construction of the left side peptide dendron **10**, terminated with orthogonally protected lysines (e.g., 2-Cl-Z residue located at *N*^α^ and Boc groups at *N*^ε^ amino groups, respectively), from the intermediate **9** was performed according to the procedure outlined in Scheme 2. Further aminolysis of the C-terminal position in **10** with 2,2′-diethylamine (DEA) or 2,2′-(ethylenedioxy)diethylamine (NOON) yielded the respective dendrons **11** and **12** (Scheme 2). Coupling with the linker required addition of high excess of the respective amines to a cooled to 0 °C methanolic solution of dendrons—100 mols for EDA and 300 mols for NOON, respectively, and further titration for five days at r.t. or for four days at 60 °C, respectively, for EDA and NOON. The resulting left side dendrons **11** and **12** equipped with the linkers were then coupled with the right side dendrons **7** and **8**. After Boc-deprotection of the amino groups with HCl/AcOEt solution, unsymmetrical bola-type molecules **16** and **22** with shorter EDA linker and **18** and **24** with longer NOON linker were obtained in the form of hexa-hydrochlorides. Analogously, the synthesis of the left side intermediate **14** terminated with four Trp residues and extended with DEA linker from the previously described compound **13** is shown in Scheme 3. Example of the synthesis of the bola dendrimer **24** from intermediate **12** is shown in Scheme 4. The protocol for preparation of the bola structures: Boc-protected bolas **19** and **25** and the respective unprotected deca-hydrochlorides **20** and **26** are presented in Scheme 5 and Scheme 6, respectively. The unprotected bola dendrimers were obtained as creamy or pale yellow hygroscopic powders with no sharp melting point.

Structure of the bola dendrimers was fully confirmed by their ESI MS and NMR spectra. In mass spectra of the protected dendrimers, main signals represent doubly ionized ions of the [M + 2Na]^2+^ structure accompanied by low intensity pseudomolecular ions of the [M + Na]^+^ type. MS spectra of the deprotected dendrimers reveal exclusively multiply charged species with most abundant [M + 3H]^3+^or [M + 4H]^4+^ ions. Due to signal overlapping in aliphatic region, ^1^H NMR spectra are of low diagnostic value. Application of DEPT technique and HSQC and COSY correlations allowed to assign signals to almost all carbon atoms in the ^13^C NMR spectra and therefore, also to the appropriate protons (Figure 1).

Full deprotection of the bola dendrimers **16**–**26** was confirmed by disappearance of the characteristic signals of the Boc group in ^1^H NMR spectra, i.e., intensive signals at 1.33–1.50 ppm [C(CH_3_)_3_], and singlets at 1.33–1.50 ppm [C(CH_3_)_3_] as well as signals at 28.7 and 28.8 ppm [C(***C***H_3_)_3_] and signals in the 79.5–81.5 ppm range [***C***(CH_3_)_3_] in^13^C NMR spectra (Figure 1). As evidenced from the NMR spectra the obtained dendrimers have at least 95% purity. Analytical data, i.e., ESI MS, ^1^H and ^13^C NMR, molar ellipticity, and melting points for the final bola dendrimers are available in Appendix A.

### 2.3. Primary Cultures of Cerebellar Granule Cells (CGC)

Primary CGC cultures were prepared from seven-day-old Wistar rats of both sexes as previously described [21,22,23]. Procedures using rat pups were performed in accordance with international standards of animal care guidelines and were approved by the Local Care of Experimental Animals Committee. The culture was incubated in growth basal Eagle’s medium (Gibco, Thermo Fisher Scientific, Grand Island, NY, USA), supplemented with 10% fetal calf serum, 25 mM KCl, 4 mM glutamine, penicillin (50 U/mL), and streptomycin (50 mg/mL) (Sigma-Aldrich, Poznan, Poland). The density of the cell suspension seeded on the 24-well plates, coated with poly-L-lysine, was 1 × 10^6^ cells per well. Cultures were treated with 7.5 µM cytosine arabinofuranoside 36 h after seeding to prevent the replication of non-neuronal cells. After seven days of culture in vitro, the cultures were used for experiments.

### 2.4. CGC Viability

Stock solutions were prepared by dissolving dendrimers in dimethylsulfoxide (DMSO) (Sigma-Aldrich, Poznan, Poland), and subsequent dilutions were made in cell medium. The effect of the dendrimers on CGC viability was assessed 24 h after a 30 min exposure of the cultures to the tested compounds using propidium iodide (PI) staining. The culture medium was replaced with Locke 25 buffer containing 134 mM NaCl, 25 mM KCl, 2.3 mM CaCl_2_, 4 mM NaHCO_3_, 5 mM HEPES (pH7.4), 5 mM glucose, and freshly prepared solutions of the tested substances in concentration: 0.2, 2, 5, 10, and 20 µM, or vehicle (0.2% DMSO), as required. After a 30 min incubation at 37 °C and washes with Locke 25 buffer, the original growth medium was replaced and CGCs were cultured for an additional 24 h. Then, the cells were fixed with 80% methanol, stained with PI (0.5 µg/mL), and viable and dead neurons were counted using an Axiovert fluorescence microscope (Carl Zeiss AG, Germany). The viability of the neurons was determined as percentages of live cells in proportion to all cells.

### 2.5. ROS Measurement

ROS production in CGCs was monitored by measuring the fluorescence of DCF, a product of the ROS-mediated cleavage and oxidation of the precursor molecule DCFH-DA (Molecular Probes, Eugene, OR, USA) that easily penetrates cells. The CGC cultures were incubated with the original culture medium containing 100 mM DCFH-DA for 30 min at 37 °C. Then, the cultures were washed three times with Locke 5 buffer containing 154 mM NaCl, 5 mM KCl, 2.3 mM CaCl_2_, 4 mM NaHCO_3_, 5 mM glucose, and 5 mM HEPES (pH 7.4). Next, the cells were incubated with Locke 5 buffer, and the fluorescence of the cell-entrapped probes was measured using a microplate reader FLUOstar Omega (Ortenberg, Germany) at 485 nm excitation and 538 nm emission wavelengths. The changes in the fluorescence intensity were recorded every 60 s. The tested dendrimers, as well as a positive control 10 µM H_2_O_2_, were added to the CGC cultures after the fifth min of the experiment. The results are presented as percent changes in fluorescence intensity relative to the basal level (F/Fo%) vs. the duration of the measurement after the addition of the test compounds.

### 2.6. Human GBM and NB Cell Lines and Human Astrocytes Cell Cultures

Human GBM T98G cell line was obtained from American Type Culture Collection (American Type Culture Collection, Manassas, VA, USA). Cells were maintained in Earle’s Minimal Essential Medium (MEME) (Sigma-Aldrich, St. Louis, MO, USA) supplemented with 10% fetal bovine serum (FBS) (Gibco), non-essential amino acids (Gibco), 50 units/mL penicillin, and 50 µg/mL streptomycin (Gibco). Human GBM LN229 cell line was a generous gift from Rafał Krętowski, PhD (Department of Pharmaceutical Biochemistry, Medical University of Białystok, Poland). Cells were maintained in Dulbecco’s Modified Eagle Medium (DMEM) (Gibco) supplemented with glucose (final concentration 4,5g/L), 10% FBS, 50 units/mL penicillin, and 50 µg/mL streptomycin. Human NB SH-SY5Y cell line was a kind gift from Anna Wilkaniec, PhD (Department of Cellular Signalling, MMRI, PAS). Cells were cultured in F12/MEM medium supplemented with 15% FBS, 1% nonessential amino acids, 50 units/mL penicillin, 50 µg/mL streptomycin, and L-glutamine. Normal human astrocytes (NHA) were purchased from ScienCell Research Laboratories (Carlsbad, CA, USA) and cultured in the Astrocyte Medium according to the manufacturer’s instruction. All cell lines were maintained at 37 °C in a humidified atmosphere with 95% air and 5% CO_2_.

### 2.7. Viability of GBM, NB, and Normal Human Astrocytes (NHA) Cell Lines Inpresence Od Dendrimers

Cells were seeded at the density of 10^5^ cells per well in a 24-well culture plate and allow to attach for 24 h. Next, cells were treated with 5 μM of the tested compounds or DMSO (the final concentration of DMSO did not exceed 0.2% *v*/*v*). After a 30 min incubation at 37 °C, medium was replaced with fresh one and cells were cultured for an additional 24 h. Then, cells were trypsinized, stained with propidium iodide (PI) (0.5 µg/mL), and counted using a TC20^TM^ Automated Cell Counter (Bio-Rad, Hercules, CA, USA). Four independent experiments were performed.

### 2.8. Colony Forming Assay

Cells were seeded in six-well plates (100 cells/well) and allow to attach for 24 h. Next, cells were treated with 5 μM of either compound **16**, **22,** or DMSO. After a 30 min incubation at 37 °C, medium was replaced with fresh one and cells were cultured for 14 days. Next, cell culture plates containing colonies were gently washed with phosphate buffered saline (PBS) (Sigma-Aldrich, Poznan, Poland) and fixed with 4% formaldehyde for 10 min. Wells were rinsed once again with PBS, colonies were stained with 0.5% crystal violet solution in 25% methanol for 10 min and the grossly visible colonies were counted. Four independent experiments were performed.

### 2.9. Statistical Analysis

Data are expressed as the mean ± SD from 3–4 independent experiments. Statistical analysis was performed using GraphPad Prism 7 (GraphPad Software). Statistical significance was determined by Student’s t-test (for comparisons between two groups). *p* < 0.05 was considered as statistically significant.

### 2.10. Antioxidant Assays

#### 2.10.1. Chemicals

2,4,6-tris(2-pyridyl)-s-triazine (TPTZ), iron(III)chloride hexahydrate, acetic acid (≥99.0%), sodium acetate, 6-hydroxy-2,5,7,8-tetramethylchromane-2-carboxylic acid (Trolox^®^), HCl, 2,2′-azino-bis(3-ethylbenzothiazoline-6-sulphonic acid)diammonium salt (ABTS), potassium persulfate, 2,2-diphenyl-1-picrylhydrazyl radical (DPPH), copper (II) chloride, ammonium acetate, and neocuproine (2,9-dimethyl-1,10-phenanthroline) were purchased from Sigma-Aldrich (Steinheim, Germany). All solvents (methanol and water) and reagents that are used in this study were HPLC or of analytical grade.

#### 2.10.2. Ferric Reducing Antioxidant Power (FRAP)

The FRAP assay was performed according to previously described method [21] with some modifications [22]. FRAP working solution was prepared freshly each time. The stock solution included 2.40 mL acetate buffer (300mM, pH 3.6), 0.24 mL 10 mM TPTZ solution in 40 mM HCl, and 0.24 mL of 20 mM FeCl_3_ × 6H_2_O in water. The mixture was kept away from light and warmed at 37 °C before using. Then, 240 μL sample Troloxor solvent (blank sample) standard solution was added to FRAP stock solution. The absorbance was measured at 593 nm after 15 min. Final concentration of methanolic solutions of the studied dendrimers were in range 0–24 μM and Trolox^®^ solutions from 0 to 50 μM.All determinations were carried out in triplicate.

#### 2.10.3. ABTS Assay

ABTS assay was based on a previously published method [23] with some modifications [24], using Trolox^®^, a water-soluble analogue of vitamin E, as standard. ABTS^•+^ radical cation was produced by soluble of 10 mg solid ABTS in mixture of water and 2.4 mM potassium persulfate aqueous solution (final concentration of ABTS 2 mM). This stock solution was kept in the dark at room temperature for 12–16 h for incubation. The ABTS^•+^ solution was then diluted with distilled water to obtain an absorbance of 1.05 ± 0.05 at 734 nm. After 240 μL sample, solvent or Trolox standard solution was added to 2.88 mL ABTS^•+^ solution; absorbance was measured at 10 min. Results of the ability of dendrimers to scavenge of cation radical ABTS are presented as Trolox equivalent antioxidant capacity (TEAC) and as IC_50_ parameter. All analyses were performed in triplicate and the results were expressed as the mean value ± standard deviation.

#### 2.10.4. DPPH Assay

Experiments were performed according to [25], with small modifications [19] 2 × 10^−3^ mol/L of DPPH radical reagent was prepared in methanol. This solution was kept in the dark at room temperature for 2 h. The DPPH^•^ solution was then diluted with methanol to obtain an absorbance of 0.95 ± 0.05 at 517 nm. Then, 240μ L of solvent, sample, or Trolox standard solutions was mixed with 2.88 mL DPPH radical solution, and after 15 min, the absorbances were measured. Results were expressed as Trolox-equivalent antioxidant capacity (TEAC) values (μM). In the concentration range investigated, 50% of radical scavenge (IC_50_) was not achieved. All experiments were repeated in triplicate, and the results were shown as the mean value ± standard deviation.

#### 2.10.5. Cupric Reducing Antioxidant Capacity (CUPRAC)

The CUPRAC method was performed according to procedure described by Apak et al. [26] with small modifications, which was necessary to correlate this method to other used assays. CUPRAC working solution was prepared freshly each time. The stock solution including 0.96 mL ammonium acetate buffer (1 M, pH 7), 0.96 mL 7.5 mM neocuproine solution in MeOH, and 0.96 mL of 10 mM CuCl_2_ in water was kept away from light. Then, 240 μL dendrimer, Trolox, or solvent (blank sample) standard solution was added to CUPRAC stock solution, and after 25 min, absorbance at 450 nm was recorded. Results were expressed as μM Trolox L^−1^. All determinations were carried out in triplicate.

## 3. Results

### 3.1. Molecular Design

Previously, we have studied a group of dendrimers bearing *p*-aminobenzoic residue (PABA) and found out that in addition to PABA an indole moiety significantly enhanced antioxidant capacity of these dendrimers, providing novel compounds that were able to significantly protect glutamate stressed rat CGC [12]. Design of the present dendrimers with nonsymmetric bola structure from two different dendrons by application of convergent methodology enabled introduction in a controlled way multiple residues with the expected cytotoxic activity: 2-chlorocarbobenzoxy groups (2-Cl-Z) or tryptophan (Trp) [13,17] and those with antioxidant properties, i.e., indole, PABA, or PAS [14] (Figure 2). Variability in dendrimer structure involved different protonation level (neutral compounds **19** and **25** vs. their (+)10 protonated derivatives **20** and **26**), different distance between left and right side dendrons provided by the linkers of different lengths (**16** containing short EDA vs. **18** containing longer NOON linker), and permutation of the essential surface groups, i.e., 2-Cl-Z vs. Trp on the left side and PABA vs. PAS on the right side dendrons (**16** vs. **22**). On the basis of the previous research that revealed positive impact of dendrimers containing antioxidant moieties on viability of neurons, it was of interest to see if such residues can modulate selectivity of dendrimers towards GBM and NB cells in comparison to the non-tumorigenic cells. Following up on previously published data that in human GBM U87 cells the cytotoxicity of four 2-Cl-Zn dendrons was based on the 2–2.5-fold increase in ROS production and the reduction of the mitochondrial membrane potential [13], all bola dendrimers designed here have been tested for their effect on the viability of primary rat cerebellar neurons (CGC) and generation of ROS. On the other hand, their molecular antioxidant activity in relation to the incorporated types of residues, in terms of redox potential (FRAP and CUPRAC tests) and free radical scavenging properties (ABTS and DPPH tests), was evaluated.

### 3.2. Impact of Bola Dendrimers on Viability of Rat Primary CGC and ROS Production

To assess the optimal dendrimer concentration providing anticipated cytotoxic activity on the malignant cells with minimal negative impact on the normal nervous system cells, viability of rat primary CGCs and ROS generation were measured. As shown in Figure 3, carrying the highest (+)10 charge dendrimer **20** containing five indole and two PABA units was the most cytotoxic. At 5 μM, it reduced viability of CGC cells by ca. 85%. On the contrary, the remaining dendrimers carrying (+)6 charge at this concentration conserved ca. 55–85% viable cells, including dendrimer **26** that had the same chemical structure and charge as dendrimer **20** except that two amide bonds on the right side dendron were replaced by the sulfonamide bonds.

Moreover, not charged dendrimer **19**, containing two PABA residues, was neutral at 5 μM concentration, and became very toxic to CGC when transformed to its (+)10 protonated form **20**. On the contrary, negative impact on CGC viability of the PAS-derivatized neutral dendrimer **25** was detected only at the highest 20 μM concentration, as compared to the protonated form **26** (Figure 3).

The studied dendrimers had relatively low impact on production of toxic radicals as compared to hydrogen peroxide (Figure 4). In general, dendrimers carrying 2-Cl-Z or Trp residues on the left side and PABA residues on the right side dendrons produced slightly more radical species than the respective analogs with PAS residues located on the right side dendron. The highest ROS amount was produced by dendrimer **16** that also expressed relatively high cytotoxicity against CGC. In case of several dendrimers, a reversed correlation between ROS production and dendrimer concentration was observed. The possible explanation might be self-aggregation of these essentially amphiphilic molecules in solution that makes them less active. All dendrimers (Figure 3) are more toxic at concentration of 20 µM. The mechanism of their toxicity is not clear, but it seems that it is not directly connected with ROS production.

### 3.3. Antioxidant Properties: Radical Scavenging Potency (DPPH and ABTS) and Redox Potential (FRAP and CuPRAC) of Bola Dendrimers

Antioxidants play multiple roles in the living systems: providing protection against the distracting effects of free radicals and securing proper oxidation level of the vital for cell functioning metal cations, e.g., Cu, Fe, and Zn, etc. In general, two types of tests are applied. The first group of antioxidant capacity assays is based on the electron transfer (ET) mechanisms. The most widely used ferric reducing antioxidant power (FRAP) and cupric ion reducing antioxidant capacity (CUPRAC) tests share similar mechanism, as both are based on the reduction of Fe(III) or Cu(II) complexes to the lower oxidation steps. However, CUPRAC test is performed at more biorelevant neutral pH, whereas FRAP test is conducted at low pH. The second group of assays involves interactions of compounds with free radicals: neutral radical scavenging assay shows capability of a compound to interact with stable neutral radicals prepared from 1,1-Diphenyl-2-picrylhydrazyl (DPPH), whereas the second test shows scavenging power against cationic radicals prepared from 2,2′-azino-bis(3-ethylbenzothiazoline-6-sulphonic acid) (ABTS). Both tests of the latter group present different aspects of antioxidative mechanism. While DPPH assay shows the capacity of antioxidant to transfer electron, ABTS assay determines cationic free radical scavenging activity involving both electron and hydrogen transfer mechanism. In both tests, the reference compound is Trolox, known radicals scavenger. Since the molecular design strategy yielded molecules with potential pro-oxidant (2-Cl-Z-substituted) and antioxidant (Trp, PABA, PAS) properties, the resulting antioxidant capacity in relation to chemical structure was determined.

The results of DPPH and ABTS tests for the (+)10-charged **16**, **18**, **20**, **22**, **24**, and **26** bola dendrimers at three concentrations (3.78, 7.69, and 15.38 μM) are shown in Figure 5, whereas results of FRAP and CUPRAC tests are shown in Figure 6. Data include also two neutral derivatives **19** and **25** for the PABA and PAS series, respectively. Considering TEAC values, dendrimers show different level of scavenging capacity in both tests, being better scavengers of radical cations than single electron bearing neutral radicals. The highest cationic radical scavenging ability in ABTS test was found for dendrimer **20** (20.3) of the PABA series, bearing five indole moieties located at the left side of the bola-dendrimer. This is probably due to stabilization of the generated dendrimeric radicals by multiple aromatic indole rings. Moreover, total cationic radical scavenging level of **20** exceeds by ca. two-fold the impact of all indole and PABA residues imprinted in the dendrimer structure (dendrimer effect). The highest neutral radical scavenging potency detected in DPPH test was found for cationic, functionalized with four 2-Cl-Z residue dendrimers **16** (5.5) and **18** (5.53) of the PABA series, which differ only by composition of a linker. On the other hand, the ROS-scavenging capability of the studied dendrimers is not much higher than the one presented by natural antioxidants. For example, caffeic acid, at relevant concentration (5.5 μM), showed values ca. 6–7 of Trolox equivalence in DPPH test [27]. To our knowledge, no antioxidant properties for PAS residue was found until present. Indeed, independent testing showed that *p*-aminobenzenesulfonic acid expressed similar scavenging potency in ABTS and DPPH tests as *p*-aminobenzoic acid (±3 e.s.d.’s), which is probably related to the presence of aromatic amino group (Figure 5). Its redox potential in terms of FRAP and CUPRAC tests is insignificant (Figure 6).

Single electron transfer phenomena could be visualized as capacity of dendrimer to reduce and stabilize metal ions at lower oxidation state, e.g., reduction of Fe(III) to Fe(II) in FRAP test or reduction of Cu(II) to Cu(I) in CUPRAC test. Significant impact on the reducing capacity of antioxidants has pH value. At acidic pH, protonation of antioxidant compounds can lower its reducing capacity. However, in the case of the studied dendrimers that in majority are already protonated at 6/10 amino groups, the effect of acidic pH should be lower. Nevertheless, markedly high reducing capacity of **16** in FRAP test (PABA series, low pH) was confirmed by conducted at neutral pH CUPRAC test. Interestingly, compound **26** of the PAS series at 3.78 and 7.69 μM concentration exhibited ca 11-fold enhancement of the antioxidant power in transition between FRAP/CUPRAC model. This property might be related to its higher chelating propensity toward the Cu^2+^ cations. Antioxidant potency of the two neutral derivatives with amino groups protected by Boc groups, i.e., **19** and **25**, is at least two times lower than that of their respective cationic derivatives **20** and **26** in all types of tests.

### 3.4. Influence of Dendrimers on the Viability and Proliferation of the Nervous System Cells

As it is shown in Figure 3, first we evaluated the effect of the selected Boc-protected (**19**, **25**) and Boc-deprotected (**16**, **18**, **20**, **22**, **24**, and **26**) dendrimers in the wide range of concentrations (0.2, 2, 5, 10, and 20 μM) on viability of rat primary neurons CGC. All tested dendrimers at the highest concentration were toxic to CGC, while at 5 μM concentration, most of them (except **20**) did not reduce CGC viability below 50–40%. Therefore, for further tests, representatives of the structurally and functionally different dendrimers were limited to a series of dendrimers **16**–**22** and the maximum concentration to 5 μM.

In this group of compounds, dendrimer **18** decreased the number of CGC neurons by 20%, while dendrimers **16** and **22** diminished cell population by approximately 40%. The most pronounced reduction in the cell number, by 80%, was observed in neurons treated with dendrimer **20** carrying five indole residues and (+)10 charge. On the other hand, the studied dendrimers decreased the number of human NB SH-SY5Y cells by approximately 40% (Figure 7A). Since dendrimer **18** presented the lowest toxicity against SH-SY5Y cells, and dendrimer **20** was extremely toxic against CGC, dendrimers **16** and **22** were subjected to a further analysis in colony-forming assay. After single 30 min treatment with dendrimers **16** or **22**, the ability of NB SH-SY5Y cells to form colonies was diminished by 50% and 80%, respectively (Figure 7B).

Contrary to the rat neurons, normal human astrocytes (NHA) turned out to be much less susceptible to the treatment with the bola dendrimers. None of the tested compounds at 5 μM concentration significantly reduced the number of NHA, as compared to DMSO-treated controls (Figure 8A). By contrast, treatment with dendrimers **16**, **18,** and **22** at 5 μM concentration diminished the number of GBM LN229 cells by approximately 30%. GBM T98G cells turned out to be less sensitive to the treatment with dendrimers, as compounds **16**, **18,** and **22** reduced the number of cells by less than 20%, while treatment with **20** gave an even lower effect (Figure 8A). Based on these results, long term effect of dendrimers **16** and **22** against GBM cells was further evaluated in the colony forming assay. Treatment of LN229 cells with dendrimer **16** reduced their ability to form colonies by approximately 50%, whereas dendrimer **22** almost completely abolished the clonogenic potential. In case of T98G cells, treatment with **16** or **22** decreased the number of colonies by 25% and 75%, respectively (Figure 8B).

## 4. Discussion

The present research was inspired by the previous observation that a group of dendrimers functionalized with several *p*-aminobenzoic residues (PABA), supplemented with an indole moiety provided novel antioxidant compounds that were able significantly protect glutamate-stressed rat CGC [12].

Design of the present dendrimers with the bola structure from two different dendrons by application of convergent methodology enabled introduction in a controlled way: multiple residues with the expected cytotoxic 2-chlorocarbobenzoxy groups (2-Cl-Z) or tryptophan (Trp)] or/and with antioxidant properties, i.e., indole system, PABA, and PAS (Figure 8). Variability in their structure involved different protonation level (neutral compound **19** vs. its deca-protonated derivative **20**), different distance between left and right side dendrons provided by the linkers of different lengths (**16** containing EDA vs. **18** containing NOON linker), and permutation of the essential surface groups, i.e., 2-Cl-Z vs. Trp on the left side and PABA vs. PAS on the right side dendrons (**16** vs. **22**).

On the basis of the previous research on positive impact of dendrimers containing antioxidant moieties on viability of neurons, it was of interest to see if such residues can modulate selectivity of dendrimers towards glioblastoma in comparison of the normal cell lines.

Functionalization of the left side dendron with Trp moiety yielded two dendrimers **20** and **26** with the highest net charge (+10). However, only combination with the second antioxidant residue, i.e., PABA (**20**) not PAS (**26**), gave negative impact on CGC viability. Termination of the dendron arms with 2-Cl-Z residue was more advantageous for the toxicity level against rat CGC and normal human astrocytes. From this group, two isostructural dendrimers **16** and **22** that differ by presence of amide or sulfonamide group in the right side dendron expressed moderate antiproliferative activity against neuroblastoma (NB) SH-SY5Y as well as glioblastoma (GBM) LN229 and T98G cell lines. In addition to limiting cell proliferation, exposition of both NB and GBM cell lines to 5 μM of dendrimers for only 30 min had a strong inhibitory effect in the colony formation test. Proliferation assay explores tumor growth potency, and colony forming assay determines the ability of a single cell to undergo “unlimited” division [28], and therefore is considered to be an indicator of undifferentiated cancer stem cells [29]. Of note, both NB and GBM contain cancer stem cells [30,31], which contribute to tumor initiation and therapeutic resistance. Thus, it is possible that dendrimers **16** and **22** influence both the mechanisms of cell division as well as the initiation of tumor and the cell resistance to treatment.

Another factor that might contribute to the selective cytotoxicity of these peptidomimetic molecules against malignant cells is their amphiphilic structure and multiple cationic character. As found by Bevers at al., mammalian cell membranes are characterized by non-symmetric distribution of phospholipids over both leaflets of the bilayer [32,33]. In particular, abundance of phosphatidylserine (PS), a phospholipid with negatively charged head, was detected in outer leaflet of a viable GBM cells. In addition to the acidic surrounding of malignant cells [34], PS became a new target for therapy of brain tumors in the following years. Recently, Riedl et al., selectively targeted negatively charged PS molecules located on the external leaflet of melanoma, GBM, and rhabdomyosarcoma as well as prostate and renal cancer cell membranes by analogs of human Lactoferricin [35,36]. This cationic and amphiphilic compound belongs to a large family of natural antimicrobial peptides (AMPs). Consequently, efficacy of various AMPs, alone [37,38,39] or as hybrid materials [40,41,42,43,44], was tested against GBM cells and showed their substantial activity along with structure-dependent versatility of mechanism of action [45,46,47]. Moreover, PS-targeting was successfully applied for the design of a new delivery systems for brain tumors imaging and therapy [48,49]. Recently, we studied interactions between 2-Cl-Z-terminated dendrimer at relatively low concentration (5.5 μM) with model lipid membranes of different fluidity by combination of neutron reflection (NR)and molecular dynamics (MD). It has been shown that upon interactions with fluid membranes (higher membrane fluidity is also typical for malignant cells), amphiphilic (+)4 charged dendrimer was inserted into the distal leaflet and caused thinning and disordering of the membrane headgroups [50]. The same model studied at longer time-scale and higher dendrimer concentration by the combination of quartz crystal microbalance (QCM-D) and atomic force microscopy (AFM) imaging during continuous flow showed significant membrane thinning due to detergent-like mechanism [51]. The presently studied dendrimers have similar overall amphiphilic character but higher positive charge and might interact with GBM cell membranes in non-specific fashion as well.

Further studies are clearly required to get insights into molecular mechanisms underlying anticancer activity of these compounds. Nevertheless, data presented above corroborate several previous notions indicating that dendrimers are promising structures displaying anticancer activity against GBM or NB cells [52,53].

## 5. Conclusions

Here, we demonstrated that the molecular basis for the selective toxicity of these nanomolecules to GBM and NB cells involves proper balance between level of cationicity and accumulation of the cytotoxic and antioxidant residues on both sides of the bola structure. Highly cationic structure, i.e., (+)10 formal charges distributed across the molecule and presence of seven antioxidant residues, most affected antioxidant potential of dendrimers in terms of cationic radicals scavenging ability (ABTS). On a cellular level, that was manifested by high cytotoxicity against normal rat and human neurons. Since all structurally different molecules generated a similar amount of radicals (ROS), their cytotoxicity against glioblastoma and neuroblastoma cells is probably not entirely related to their antioxidant character. The studied polyvalent dendrimeric molecules with potential antioxidant/prooxidant behavior unexpectedly governed not only proliferation, but also long term colony formation propensity.

## Data Availability

Analytical data, i.e., ESI MS, ^1^H and ^13^C NMR, molar ellipticity, and melting points for the final bola dendrimers are available in Appendix A.

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
