# Peer review of "Peptide Dendrimers with Non-Symmetric Bola Structure Exert Long Term Effect on Glioblastoma and Neuroblastoma Cell Lines"

_biomolecules, 2021, doi:10.3390/biom11030435_

Round 1

Reviewer 1 Report

very nice paper, well written. May be a list of abbreviations could be helpful. Just a question: Compounds 19 and 25 contain no charge, contrarily to the other compounds; are they as soulble in water (or serum) than the others?

Author Response

Very nice paper, well written. May be a list of abbreviations could be helpful. Just a question: Compounds 19 and 25 contain no charge, contrarily to the other compounds; are they as soulble in water (or serum) than the others? 

Thank you. A list of “Abbreviations” has been added just after “Keywords” section. As regards dendrimers solubility issue, initial stock solutions were prepared by dissolving dendrimers in dimethylsulfoxide (DMSO) and subsequent dilutions were made in cell medium, keeping DMSO content below 0.05 %. Solutions were always clear and no precipitation was observed. For non-biological testing both cationic and neutral derivatives were dissolved in MeOH, that is very good solvent for the majority of our peptide dendrimers.

Reviewer 2 Report

The data exposed is very interesting, manuscript is also well-written and easy to read. However, some details are missing like:

  1. A small diagram with the different dendrimers tested in cells is missing. There is one at the very end of the manuscript but it would be easy to follow it together with the biological assay.
  2. More discussion concern dendrimer toxicity mechanism is missing. Discussion is in general poor.
  3. It is not clear why dendrimers would be more toxic for cancer cells rather than normal cells.

Author Response

The data exposed is very interesting, manuscript is also well-written and easy to read. However, some details are missing like:

  1. A small diagram with the different dendrimers tested in cells is missing. There is one at the very end of the manuscript but it would be easy to follow it together with the biological assay.

Thank you very much. Indeed, new Figure is necessary to follow discussion on structural aspects of the designed dendrimers. Therefore, previous Figure 8 was transferred from “Discussion” to the “Molecular design” section and supplemented with three more representative compounds that are discussed in the text (now Fig. 2). Consequently, numbering of the consecutive Figures has been modified in figure captions and in the text. 

  1. More discussion concern dendrimer toxicity mechanism is missing. Discussion is in general poor.

We addressed this issue in a new large paragraph of the “Discussion” section. It concerns the known fact of non-symmetric distribution of certain phospholipids, e.g. phosphatidylserine (PS) on both leaflets of the viable cancer cell membranes (Bevers et al., 1996). PS has negatively charged head what made it new target in cancer diagnostics and therapy. This approach was successfully applied by Riedl et al., 2011; and others for designing a group of linear peptides originated from cationic, apmphiphilic natural peptides (known as AMP’s). These were proposed for use in treatment of glioblastoma, melanoma, and other cancer types. Our branched peptide molecules are peptidomimetics of AMP’s and share with them structural features like amphiphilicity and polycationic character. Mechanism of biological activity of these branched peptides is a major motif of our research for many years. In particular, we studied compounds with dendrimer arms terminated with 2-Cl-Z residue using wide range of physicochemical methods (analogous to compounds 166, 18, 22, etc.). We are now referring to these studies in the revised version of the manuscript.

  1. It is not clear why dendrimers would be more toxic for cancer cells rather than normal cells.

The acidic surrounding of malignant cells and the above mentioned external overexpression of PS in viable cancer cells is significant source of selectivity of our polycationic dendrimers towards GB cells. We hope that the extensive discussion of the above issues along with addition of a new references (ref. 32 – 51) will now enhance value of the manuscript.